# Validation of Novel Reference Genes in Different Rice Plant Tissues through Mining RNA-Seq Datasets

**DOI:** 10.3390/plants12233946

**Published:** 2023-11-23

**Authors:** Xin Liu, Yingbo Gao, Xinyi Zhao, Xiaoxiang Zhang, Linli Ben, Zongliang Li, Guichun Dong, Juan Zhou, Jianye Huang, Youli Yao

**Affiliations:** 1Jiangsu Key Laboratory of Crop Genetics and Physiology/Co-Innovation Center for Modern Production Technology of Grain Crops, Yangzhou University, Yangzhou 225009, China; liuxinkiq@163.com (X.L.); yingbogao_yzu@163.com (Y.G.); xyzhao_o@163.com (X.Z.); m18752788661@163.com (L.B.); lzl4737@163.com (Z.L.); gcdong@yzu.edu.cn (G.D.); juanzhou@yzu.edu.cn (J.Z.); 2Lixiahe Agricultural Research Institute of Jiangsu Province, Yangzhou 225007, China; zhngyz@126.com

**Keywords:** reference genes, RNA-seq datasets, *Oryza sativa*, RT-qPCR, gene expression

## Abstract

Reverse transcription quantitative real-time PCR (RT-qPCR) is arguably the most prevalent and accurate quantitative gene expression analysis. However, selection of reliable reference genes for RT-qPCR in rice (*Oryza sativa*) is still limited, especially for a specific tissue type or growth condition. In this study, we took the advantage of our RNA-seq datasets encompassing data from five rice varieties with diverse treatment conditions, identified 12 novel candidate reference genes, and conducted rigorous evaluations of their suitability across typical rice tissues. Comprehensive analysis of the leaves, shoots, and roots of two rice seedlings subjected to salt (30 mmol/L NaCl) and drought (air-dry) stresses have revealed that *OsMED7*, *OsACT1*, and *OsOS-9* were the robust reference genes for leaf samples, while *OsACT1*, *OsZOS3-23*, and *OsGDCP* were recommended for shoots and *OsMED7*, *OsOS-9*, and *OsGDCP* were the most reliable reference genes for roots. Comparison results produced by different sets of reference genes revealed that all these newly recommended reference genes displayed less variation than previous commonly used references genes under the experiment conditions. Thus, selecting appropriate reference genes from RNA-seq datasets leads to identification of reference genes suitable for respective rice tissues under drought and salt stress. The findings offer valuable insights for refining the screening of candidate reference genes under diverse conditions through the RNA-seq database. This refinement serves to improve the accuracy of gene expression in rice under similar conditions.

## 1. Introduction

Gene expression analysis is a critical approach to exploring functions of target genes in molecular research. Since gene expression is fundamentally regulated at transcription level, studies often need to elaborate the expression level of mRNA of a target gene. Common techniques for measuring gene expression include: Northern blot, in situ hybridization, reverse transcription polymerase chain reaction (PCR), microarray, and RNA-sequence (RNA-seq). Among them, reverse transcription quantitative real-time PCR (RT-qPCR) is much more frequently used for quantifying the mRNA levels of specific genes thanks to its specificity, sensitivity, flexibility, scalability, and high-throughput [1,2]. Quantification of RT-qPCR products are generally indicated and calculated by the relative values of cycle numbers to certain chosen stably expressed genes reference genes [3]. Unless quite stably expressed genes are chosen as reference, the expression level in a specific sample is often overestimated or underestimated, and differential expressions are prone to errors. A specific gene of research interest is likely displaying a contrasting expression level across tissues, especially when different reference genes are used. For example, when *OsUBQ* [4] and *OsACTIN* [5] are separately used as a reference gene in the same laboratory, their same target gene (LOC_Os01g12890) displays a contrasting ratio of (1;0.6;1.4;0.6) vs (1;3;10;6) in roots, stems, leaves, and young panicles, respectively. Apparently, in addition to inconsistency in their probable growth condition, choosing a different reference gene likely contributes to the inconsistency in their results. Obviously, choosing an unsuitable reference gene in gene expression assays may lead to confounding results and confusing conclusions [6]. The validity of a reference gene is critical for generating reliable and accurate RT-qPCR results [7,8]. In a vast gene expression analysis, a variety of genes are generally considered to be reference genes in rice plants, such as UBIQUITIN (*UBQ*), ACTIN (*ACT*), TUBULIN (*TUB*), GLYCERALDEHYDE-3-PHOSPHATE DEHYDROGENASE (*GAPDH*), TYPE 2 C PROTEIN PHOSPHATASES (*PP2C*), ELONGATION FACTOR (*EF*), F-BOX DOMAIN PROTEIN (*F-box*) and NAC TRANSCRIPTION FACTOR (*NAC*) genes [9,10,11,12,13]. However, as studies accumulate, it has been demonstrated that the transcript levels of these genes are responsive to multiple common growth factors and vary substantially in development stages [9,14,15]. *UBQ* has traditionally been regarded as a reliable reference gene across various species. However, its expression stability in sugarcane under drought conditions has been proven to be inconsistent [16]. Similarly, *ACTIN* exhibits instability at different stages of grape development [17]. As for rice, *Osβ-TUB4* (LOC_Os01g59150) and *OsGAPDH* (LOC_Os04g40950) emerge as the least stable candidate reference genes for expression in various organs and different stages of rice plants [15]; *OsUBQ10* (LOC_Os06g46770) and *OsGAPDH* (LOC_Os04g40950) exhibit the lowest stability rankings in a 7-day-old rice seedlings treated with phytohormones and stress conditions [9]. These evidently manifest the need to refine the screening of reference genes and identify genes with higher expression stability [18,19].

RNA-seq data provide us a comprehensive and accurate global expression profile of every active gene in the genome. The use of RNA-seq data has been demonstrated to be a highly reliable method for selecting reference genes in rice. By digging into RNA-seq and microarray data of developing endosperms, *OsFb15*, *OseIF-4a*, and *OsUBQ5* are proved to be stable reference genes under normal or high temperature conditions [20]. Screening from publicly available transcriptomic datasets exposed to heavy metal stress identifies *OsOBP* and *OsCK1a.3* to be the most reliable reference genes under similar conditions [12]. Nonetheless, their selection of RNA-seq datasets was limited to specific treatment scenarios. To identify reference genes with a broader applicability, we conducted a screening of RNA-seq datasets encompassing diverse varieties, organs, and environmental conditions. Before our study, other researchers had employed public transcriptome databases to identify stable reference genes for rice [15,21]. However, when we employed our own database for screening, we found that all of these reference genes exceeded the range of 0.75 to 1.25 (FPKM/FPKM average) in one or more RNA-seq datasets, with a majority of them surpassing this range in the case of salt-stressed RNA-seq data. This underscores the notion that the reference genes identified in prior studies may not be universally applicable and indirectly validates the robustness of our screening method as well as the comprehensiveness of our RNA-seq dataset. Taking the advantage of our accumulated multiple RNA-seq datasets, we screened out 12 genes with relatively stable expression as candidate reference genes (*OsACT1*, *OsF-Box1*, *OsPP2C1*, *OsPP2C2*, *OsUBQ1*, *OsUBQ2*, G-PATCH DOMAIN CONTAINING PROTEIN (*OsGDCP*), HELICASE FAMILY PROTEIN (*OsHeFP*), COMPONENT OF THE MIDDLE MODULE OF THE MEDIATOR COMPLEX (*OsMED7*), HOMOLOGUES OF YEAST YOS-9P (OsOS-9), RETROTRANSPOSON PROTEIN (*OsReTP*), and ZOS3-23-C2H2 ZINC FINGER PROTEIN (*OsZOS3-23*)). Yet, to avoid inconsistency in gene expression of the target genes due to suboptimal performance of endogenous control, it is necessary to validate these candidate genes [22]. Different statistical algorithms and methods such as geNorm ver. 3.5 [23], NormFinder ver. 0.953 [24], BestKeeper ver. 1.0 [25], the comparative ΔCt [26], and RefFinder [27] have been developed to evaluate the best suited reference genes for normalization of RT-qPCR data in a given set of samples.

Moreover, RiceXPro (Rice Expression Profile Database; http://ricexpro.dna.affrc.go.jp/; accessed on 1 October 2022) databases provide access to curated microarray analysis data [28]. RiceXPro is a comprehensive platform that contains gene expression profiles under different growth stages and experimental conditions, which are grouped into three categories: field/development, plant hormone, and cell and tissue type. In order to identify reference genes with broader applicability across different scenarios, we commenced by analyzing the stability of reference genes that have been commonly used in previous research. Subsequently, we screened 12 candidate reference genes using six RNA-seq datasets, which encompassed a diverse range of rice varieties, tissues, and treatment conditions. To further assessing their stability, we utilized both the RNA-seq dataset and the RiceXPro database. Finally, to confirm their reliability, we used the candidate reference genes to validate the expression patterns of three target genes in the leaf, shoot, and root tissues under salt and drought stress conditions: *OsbZIP71,* which is responsive to salt and drought [29]; *OskTN80b*, which is non-responsive to salt and drought [30], and a nitrate transporter *OsNRT1.1B*, which is assumingly not closely related to these stress factors [31].

## 2. Results

### 2.1. Validation of Already Reported Common Reference Genes

Based on rice research articles from a public database, we summarized eight existing reference genes that are commonly used: OsUBQ5 (LOC_Os06g44080), OsGAPDH-1 (LOC_Os04g40950), Osβ-TUB (LOC_Os01g59150), OseIF4α (LOC_Os02g12840) [9], OsACT11 (LOC_Os03g50885) [13], UBIQUITIN-CONJUGATING ENZYME 32 (OsUBC32) (LOC_Os02g42314) [10], OsGAPDH-2 (LOC_Os04g51690) [11], and CASEIN KINASE 1a.3 (OsCK1a.3) (LOC_Os01g56580) [12]. These eight commonly adopted reference genes have been used in multiple rice tissues, such as those exposed to drought stress, diseases, iron toxicity, and heavy metal stress. To check their stability, we examined their FPKM values in our RNA-seq datasets. Among them, the FPKM values of OseIF4α was low, indicating limited abundance in expression; and OsACT11, OsGAPDH-1, and Osβ-TUB had high FPKM values, indicating rich abundance, yet, with a wide variation range (Figure 1a). Among them, only OsCK1a.3, OsGAPDH-2, and OsUBC32 have relatively stable expression abundances, although there are some outliers in the FPKM values of OsCK1a.3 and OsUBC32.

RiceXPro (Rice Expression Profile Database; http://ricexpro.dna.affrc.go.jp/) provides access to curated microarray analysis data. We utilized this database to examine the stability of these eight existing reference genes in the various organs and tissues of rice in different growth stages and constructed a heatmap of these eight genes at different developmental stages (Figure 1b). Results from RiceXPro did corroborate the RNA-seq results. The transcript levels of OsACT11, OseIF4α, and Osβ-TUB showed higher fluctuation with log2 differences greater than four (transcript level at 16 times); the log2 differences of OsGAPDH-1 were greater than two (four times at transcript level). While there are outliers in the FPKM values of OsCK1a.3 and OsUBC32, their data in RiceXPro still exhibit relative stability. In conclusion, most of these existing reference genes were not as stable as they are assumed to be. Therefore, it appears necessary to find more suitable reference genes from the existing RNA-seq datasets.

### 2.2. Screening for Stably Expressed Reference Genes from RNA-Seq Datasets

In our quest for more universally applicable candidate reference genes, we screened our existing RNA-seq datasets, which comprise responses of two *japonica* rice varieties to salt stress, one *japonica* and one *indica* rice variety to three nitrogen application rate [32], one *japonica* and one *indica* rice variety to γ-irradiation dosage [33], and a *japonica* rice variety to carbon dioxide concentration and nitrogen rate [34] treatments. These datasets include samples from root, shoot, 7-day-young whole plants, filling grain at seven days post anthesis, and shoot tip growth points of rice, containing RNA-seq results from a total of 72 biological samples of various tissues. These RNA-seq datasets encompass a range of conditions, including non-stress normal growth conditions, abiotic stress, and extreme growth conditions. The goal was to identify candidate reference genes with broad applicability. After subjecting the FPKM values of genes in these datasets to an ANOVA test, 2776 genes were found showing non-significant change (*p* > 0.05) across different varieties and growth conditions. Genes that presented at least twice in the six databases were selected (totally 89 genes), and their expression data were collected in the Genevestigator. Pearson correlation analysis on the expression levels of these genes across various tissues identified 12 genes with the highest positive correlation coefficients (R^2^ > 0.95, *p* < 0.01). Among them, six genes (*OsACT1*, *OsF-Box1*, *OsPP2C1*, *OsPP2C2*, *OsUBQ1* and *OsUBQ2*) belong to the commonly adopted reference gene families, but are not widely used, while the other six genes (*OsGDCP*, *OsHeFP*, *OsMED7*, *OsOS-9*, *OsReTP* and *OsZOS3-23*)) belong to new groups (Table 1). Following a comprehensive data screening process, we observed that the ratio of FPKM values to the average values of commonly used reference genes, *OsCK1a.3*, *OsGAPDH-2*, and *OsUBC32*, in the RNA-seq dataset of salt stress conditions fell outside the expected range of 0.75–1.25. Apparently, these three genes demonstrated limited stability and failed to be reference genes in a broader treatment condition.

To validate the applicability of these new candidate reference genes in a wet laboratory, we designed RT-qPCR primers by using the NCBI Primer-BLAST web-based tools (see methods section for details) and tested the amplification efficiency of these primers in actual RT-qPCR analysis (Table 1). The amplification efficiencies ranged from 91.08% to 108.05%, the linear regression correlation coefficients (R^2^) for all 12 genes were ≥0.98, and all exhibited suitable ranges [35]. Therefore, these 12 genes were selected as novel candidate reference genes for further stability validation.

### 2.3. Expression Profiling of the Novel Candidate Reference Genes

In quantitative PCR, the Cq value is a scale indicating transcript abundance, with a smaller Cq value reflecting a higher template number (transcript level), and vice versa. The mean Cq values of the 12 candidate reference genes in trial samples varied from 24.19 to 39.99, with the least variations observed for *OsACT1* and *OsReTP* (Figure 2a).

In parallel, we compared the Log_2_(FPKM) values from the RNA-seq datasets for these candidate reference genes (Figure 2b). Overall, the FPKM value indicated that their expression levels were all less variable than previous commonly used reference genes (Figure 2b vs. Figure 1a).

The expression profiles of these 12 candidates at different developmental stages from RiceXPro were also shown in a heatmap. All these novel candidates showed a relatively more stable transcript level (Figure 2c) compared with previous commonly used reference genes (Figure 1b). These indicate the novel candidate reference genes vary less in their expression levels.

### 2.4. Expression Stability of the Novel Candidate Reference Genes Based on Five Different Statistical Algorithms

To further compare their relative suitability as a reference gene, we used five common statistical algorithms to quantify their differences.

geNorm analysis ranks the candidate reference genes according to their average expression stability (*M* value) by using the Cq value of all samples. This evaluation is biased more on the abundance of a candidate. Genes with the lowest *M* values are considered the most stable ones. Based on the geNorm analysis, *M*-values were calculated for leaf, shoot, and root subjected to different treatments, respectively. The *M* ranking of the reference genes apparently differed in various tissues. The lowest *M*-values (most stable) were *OsMED7* and *OsACT1* in the leaf tissue, *OsZOS3-23* and *OsGDCP* in the shoots, and *OsMED7* and *OsOS-9* in the roots (Figure 3a–c). On the contrary, higher *M*-values (unstable) were observed for *OsReTP* and *OsF-Box1* in most leaf samples, *OsUBQ2* and *OsPP2C1* in shoot samples, and *OsPP2C2* and *OsPP2C1* in the roots. These suggest that it is better to choose the proper reference gene according to the tissue type of detection.

NormFinder evaluates the stability of all candidate reference genes by their intra-group and inter-group variations, with lower values indicating more stability. In the trial samples of different tissues, *OsMED7* and *OsACT1* were shown to be the most stable references in both leaves and shoots, yet with a different order; whereas *OsOS-9* and *OsMED7* appeared the most stable in the roots (Figure 3d–f). The least stable ones were *OsReTP* and *OsF-Box1* in leaf samples, *OsUBQ2* and *OsPP2C1* in shoots, and *OsPP2C2* and *OsPP2C1* in root samples. The list of stable reference genes screened out in the leaf and root by NormFinder was basically consistent with those produced by using the geNorm algorithm but with a different one in shoot tissue.

BestKeeper determines the stability of a reference gene by the extent of SD, with a higher SD value referring to low stability of the housekeeping genes and vice versa. Results from the trial samples revealed that the most stable reference genes were *OsACT1* and *OsMED7* in both leaf and shoot samples and *OsZOS3-23* and *OsGDCP* in the roots (Figure 3g–i). The least stable reference genes were *OsF-Box1* and *OsReTP* in leaves, *OsUBQ2* and *OsPP2C1* in shoots, and *OsReTP* and *OsPP2C1* in the roots. The stable ones with this algorithm were the same as with the previous two methods in the leaf, with results being very different in the root or shoot.

The ΔCt method defines the stability of housekeeping genes by using comparative ΔCt based on standard deviation (SD). The higher the SD value is, the lower the stability of a reference gene, and vice versa. By using this approach, *OsMED7* and *OsACT1* were found to be the most stable genes for the leaf, *OsACT1* and *OsZOS3-23* were the most stable for the shoots, and *OsMED7* and *OsGDCP* were the most stable for the roots (Figure 3j–l). Apparently, the list of the most stable references found by using this method was consistent with the previous algorithms only for the leaf.

RefFinder calculates the comprehensive ranking of a candidate reference gene by combining multiple analysis programs (geNorm, NormFinder, BestKeeper). Results from RefFinder showed that the top three stable reference genes were *OsMED7*, *OsACT1*, and *OsOS-9* in leaf samples, *OsACT1*, *OsZOS3-23*, and *OsGDCP* in the shoot, and *OsMED7*, *OsOS-9*, and *OsGDCP* in the root (Figure 3m–o). The least stable reference genes were *OsReTP* and *OsF-Box1* for leaf samples, *OsUBQ2* and *OsPP2C1* for shoot samples, and *OsPP2C2* and *OsPP2C1* for root samples, respectively.

Different approaches generated inconsistent results regarding the stability of candidate reference genes, indicating the limitation of each algorithm, and suggesting flexible adoption of different reference genes in various tissues under specific growth conditions. Since RefFinder produces a comprehensive ranking of the candidate reference genes, and the top list from this method overlaps the most with the other four algorithms, we choose to use the three top-ranked genes to compare their performances in actual PCR quantification analysis. For comparison purposes, two candidate reference genes in the bottom rank of each tissue type were used as negative controls for gene expression normalization.

### 2.5. Expression Profiling of Target Genes by Using Different Novel Candidate Reference Genes

The three target genes tested in the trials are: *OsbZIP71*, which is sensitive to salt and drought [29], *OskTN80b*, which is non-responsive to salt and drought [30], and nitrate transporter *OsNRT1.1B* as a by-stander [31]. The results are shown in Figure 4.

It was evident that the leaf expression levels and patterns of *OsbZIP71*, *OskTN80b*, and *OsNRT1.1B* in response to drought and salt treatments were generally similar, either normalized singularly by *OsMED7*, *OsACT1*, or *OsOS-9* or by the three combined (*OsMED7* + *OsACT1* + *OsOS-9*). Yet, it was obviously different from those normalized by *OsReTP* or *OsF-Box1*. Similar consistency and differences were also noticeable in the shoots, and roots, respectively, by using our newly selected proper reference gene or by using the least preferred ones. This indicated that choosing one proper reference gene to calculate the expression level of a target gene did not compromise reflecting the response pattern to a treatment factor, compared with using three reference genes combined. Therefore, these results validated the reliability of the these newly selected candidate reference genes. It is interesting to note that even though *OskTN80b* is claimed to be non-responsive to drought and salt stress, its expression responded similarly with a confirmed responsive gene *OsbZIP7*, though at a much lower scale.

## 3. Discussion

Though RT-qPCR is a fast, reliable, and sensitive technique for quantifying the transcripts of an expressing gene of interest, it depends on the normalization procedures by using suitable reference genes to minimize the variation in sample preparation and reactions [36]. Ideally, a reference gene should be expressing constantly, with a minimal variation in transcript level, irrespective of tissue type or experimental treatment [37]. However, accumulated studies have revealed that the expression of most adopted reference genes undergoes significant changes in different organs [15,38] across various growth stages [39] and cultivation conditions, even in a specific variety of the same species [16]. Probably there are no perfectly stable expressing reference genes at all. Unless we fully adopt digital PCR [40] to obtain an absolute quantification of the templates, we must compromise by choosing proper reference genes in RT-qPCR.

On the other hand, rice is a vital crop with substantial socio-economic significance. It serves as the staple food for more than 50% of the world’s population. Furthermore, it stands as a model crop for scientific research [41,42], and numerous studies have been conducted on the identification of reference genes in rice [9,10,12,13,15,20]. Nonetheless, these studies frequently exhibit limitations related to growth conditions, rice varieties, and rice organs.

Along with the sharp cost reduction in RNA-seq [43], the booming accumulation of transcriptomic data has generated accurate and comprehensive profiles of global gene expression in different species, various organs, across development phases, and growth conditions, which provide a great opportunity to revisit choosing a proper reference gene [14,36]. Here, by digging into our six existing RNA-seq datasets of 72 independent biological samples from rice plants, we screened out 12 new candidate reference genes with relatively more stable expression. Their FPKM values were greater than 70% of the expressing genes overall and showed significant correlation in expression level across a variety of rice cultivars, tissue types, growth stages, and treatment conditions. Quantification in test samples proved that they belong to the abundantly expressing group. Extraction data of these locus from RiceXPro database show that they vary less than most of those previous commonly adopted reference genes. Statistical analysis by using the established reference gene appraisal methods revealed that ranking of the newly selected reference genes was generally consistent for the top two to four candidates, but with a distinct variation among tissues. Quantification of evaluation indicated that for the leaf samples in rice, *OsMED7*, *OsACT1*, and *OsOS-9* were the most reliable reference genes, while for the shoot samples, *OsACT1*, *OsZOS3-23*, and *OsGDCP* were recommended; as for the root samples, *OsMED7*, *OsOS-9*, and *OsGDCP* were the best choices. Additionally, we harnessed the web-based edition of LinRegPCR (https://www.gear-genomics.com/rdml-tools/; accessed on 30 October 2023) for in-depth analysis of the RT-qPCR data, leading to quality assessments grounded in amplification and melting curve data [44]. Our findings confirmed that the individual PCR efficiency for all genes employed in quantification exceeded 1.7.

It has been proposed as a golden rule of RT-qPCR to include at least four reference genes to reduce the deviation by a single reference gene [6]. However, since including more reference genes brings in extra tedious lab work and a higher cost of chemicals and labor, most researchers prefer a singleton. Our results in validation quantification of target genes using novel reference genes (Figure 4) showed that with the choice of a singleton from these priority lists, the relative expression quantification of the target gene was literally consistent. A correct choice of reference gene will not only provide accurate assessment of the transcript responses of the gene of interest but also can save lab costs and labor time as well.

Nevertheless, despite the stability of the reference genes identified in this study across diverse parameters such as development, tissue, and various stresses in six RNA-seq datasets, it is essential to validate the stability of the respective choice of the selected internal reference genes [19]. As more transcriptomic data accumulates, it will be interesting to collect as many RNA-seq datasets as possible and provide a more comprehensive search for specific optimal reference genes by species/cultivar, tissue type, developmental stage, growth condition, and biotic/abiotic stress treatment. This may help researchers in swiftly tailoring their appropriate reference genes for their specific species, tissues, or experimental conditions.

## 4. Materials and Methods

### 4.1. Plant Material and Treatments

To further validate the stability of candidate reference genes, two *japonica* rice cultivars were chosen for conducting subsequent RT-qPCR experiments. Seeds were sterilized and imbibed in distilled water at a 4 °C refrigerator for three days before being subjected to germination at 32 °C in an incubator for three days under a 12/12 h of light/dark cycle. Seedlings with similar shoot length were transferred to grow hydroponically (in water) for seven days at 28 °C and the same light/dark cycle in a growth chamber at an illumination of 150 μmol m^−2^·s^−1^. For salt stress treatment, the hydroponic solution was subsequently replaced with 30 mmol/L NaCl; for the drought treatment, the seedlings were subjected to air-drying; and mock treatment with a water served as a control. The leaves, shoots, and roots were collected thirty minutes after treatment, promptly frozen in liquid nitrogen, and stored at −80 °C before RNA extraction.

### 4.2. RNA Isolation and cDNA Synthesis

RNA was extracted using the RNA Easy Fast Plant Tissue Kit and any genomic DNA residue was removed using RNase-free DNase I per the manufacturer’s instructions (Tiangen, Beijing, China). Concentration and purity of the RNA samples were measured with the spectrophotometer NanoDrop 2000 (Thermo Fisher Scientific, Wilmington, DE, USA). In accordance with the manufacturer’s instructions, 1.0 μg of total RNA was reverse-transcribed using a 20 μL volume of a Fast Quant RT Kit (Tiangen, Beijing, China).

### 4.3. Selection of Candidate Reference Genes and Design of Primers 

Candidate reference genes were screened and selected according to the bioinformatic analysis of six RNA-seq datasets from diverse organs of different rice varieties. These data were generated from varieties including Nipponbare (*japonica*), 9311 (Yangdao 6, *indica*), Wuyunjing 8 (*japonica*), Wuyunjing 30 (*japonica*), and Zhen23309 (*japonica*); tissue types including germinating shoot, germinating root, shoot tip with ~50 mm of the growth point, whole young shoot, young root, and filling grain at seven days post anthesis; and treatment factors including nitrogen rates, atmospheric CO_2_ levels, γ-irradiation, and NaCl stresses. Selection of candidate reference genes was conducted by using the following criteria: (i) the gene must present at a high level in all tissues across all treatments and data sets included in the analysis [that is, at least one of its protein-coding transcripts must have an expression level higher than 70% of the whole sample’s average fragments per kilobase million (FPKM)]; (ii) the ratio between the individual treatments’ FPKM and the global average FPKM of the transcript in the specific data set [(individual FPKM)/(average FPKM)] must be in the range 0.75–1.25; (iii) the variability of transcript expression should be low within all tissues and across all treatments as evidenced by the *p* value of ANOVA >0.05. After the screening, we selected genes that were often represented in many datasets, examined their expressions in various rice tissues using Genevestigator (Immunai/NEBION AG, Zurich, Switzerland), and selected candidate reference genes based on their ranking by correlation coefficient (R^2^) using correlation analysis (SPSS25, IBM, Chicago, IL, USA).

The primers for the twelve candidate reference genes and three target genes were generated using the NCBI Primer-BLAST web-based tools (https://blast.ncbi.nlm.nih.gov/Blast.cgi; accessed on 1 October 2022) with the following parameters: (1) a GC content of 40–60%; (2) an amplicon size of 78–133 base pairs (bp); and (3) a melting temperature (Tm) of 56.3–61.2 °C. Table 1 lists the primer sequences, amplicon length, and Tm of twelve candidate reference genes and three target genes. Primers were purchased from Qingke Company Ltd. (Nanjing, China).

### 4.4. Quantitative PCR Conditions

RT-qPCR was performed in 96-well plates using Bio-Rad CFX Real-Time PCR equipment with a SYBR green-based PCR system (Bio-Rad, Hercules, CA, USA). Each reaction had a final volume of 12 μL and contained the following components: 1 μL cDNA, 6 μL SYBR Green (Bio-Rad, Hercules, CA, USA), 0.5 μL forward primer (10 μmol/L), 0.5 μL reverse primer (10 μmol/L), and 4 μL ddH2O. The reaction was conducted at 98 °C for 1.5 min, followed by 44 cycles of denaturation at 95 °C for 5 s, annealing at 56.3–61.2 °C for 5 s, and extension at 72 °C for 10 s. After 44 cycles, the dissociation curve profiles were examined to determine the specificity of the amplicons (melting curve). Each RT-qPCR involved three biological replicates and three technical replicates.

### 4.5. Data Analysis

Bio-Rad CFX Manager software (Bio-Rad, Hercules, CA, USA) was used for the calculation of the Cq values of each amplification. The expression stability of each candidate reference gene was assessed using five distinct methodologies: geNorm ver. 3.5 [19], NormFinder ver. 0.953 [20], BestKeeper ver. 1.0 [21], the comparative Ct [22], and RefFinder [23] (https://blooge.cn/RefFinder/; accessed on 1 October 2022) and the method therein. geNorm computes the *M* value to indicate the stability of a reference gene; the greater the stability of a reference gene, the smaller the *M* value is. NormFinder uses an ANOVA-based model to check the expression stability of a reference gene by evaluating inter- and intra-group variation; the gene with the lowest value has the most stable expression. BestKeeper estimates the stability of gene expression by using standard deviation (SD), a lower SD indicates greater stability of the reference gene. The Ct method compares the relative expression of all pairwise combinations of candidate reference genes to determine which gene has the most stable expression by calculating the standard deviation of the pair’s relative expression; the lower the average standard deviation, the more stable the expression of a candidate reference gene. RefFinder combines multiple algorithms to identify the most appropriate reference gene.

## 5. Conclusions

Twelve novel candidate reference genes from multiple RNA-seq datasets were mined out, and their applicability was evaluated in different tissues of the rice plant (*Oryza sativa*). Our findings indicate that *OsMED7*, *OsACT1*, and *OsOS-9* emerged as the most robust reference genes for leaf samples, while *OsACT1*, *OsZOS3-23*, and *OsGDCP* were the most suitable for shoots and *OsMED7*, *OsOS-9*, and *OsGDCP* were the most reliable reference genes for roots. A singleton of these reference genes can be utilized in subsequent experiments in different rice tissues under drought or salt stress conditions to ensure accurately assessing expression of a target gene. Meanwhile, our research suggests that the RNA-seq analysis method holds great potential for the discovery of new, stable reference genes, which will contribute to assessing gene expression levels not only in rice but also in other species.

## Figures and Tables

**Figure 1 plants-12-03946-f001:**
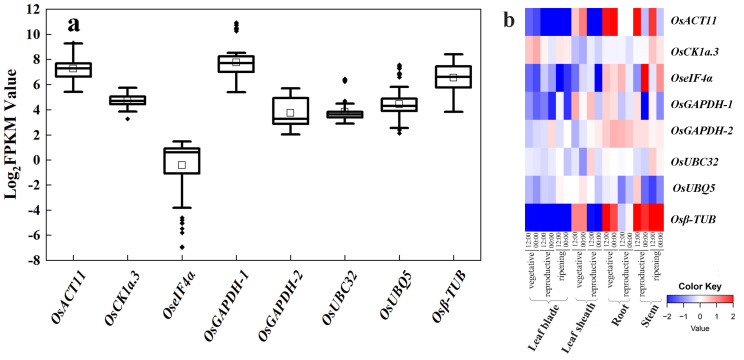
Expression profile of eight commonly used reference genes. (**a**) Expression displayed as Log_2_(FPKM) values for eight reference genes (*OsACT11*, *OsCK1a-3*, *OseIF4α*, *OsGAPDH-1*, *OsGAPDH-2*, *OsUBC32*, *OsUBQ5*, and *Osβ-TUB*) in the six RNA-seq datasets. The horizontal line in the box is the median line, the white box is the mean, and the black dot outside the box is the outlier. (**b**) Heatmap of expression of eight commonly used reference genes, created using data from RiceXPro (including different developmental stages and circadian time points). Blue indicates low transcript levels, and red indicates high transcript levels in log2 values.

**Figure 2 plants-12-03946-f002:**
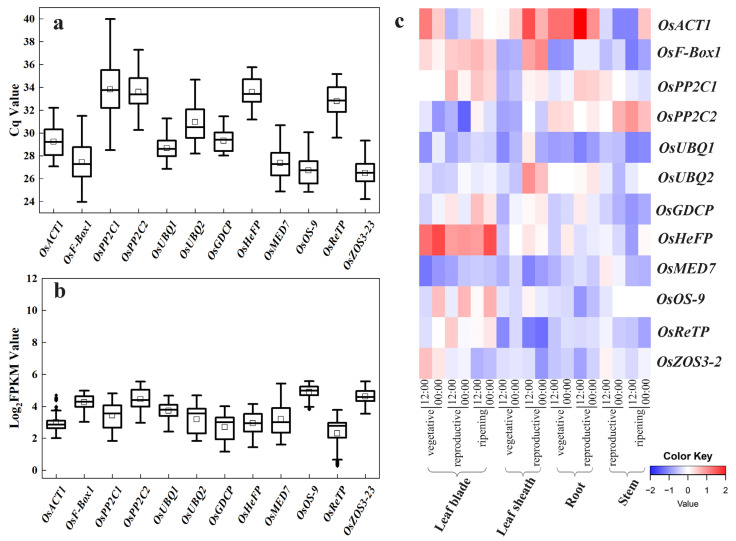
Expression profiling of the 12 candidate reference genes (*OsACT1*, *OsF-Box1*, *OsPP2C1*, *OsPP2C2*, *OsUBQ1*, *OsUBQ2*, *OsGDCP*, *OsHeFP*, *OsMED7*, *OsOS-9*, *OsReTP*, and *OsZOS3-23*). (**a**) The Cq values of the 12 candidate reference genes across the leaf, shoot, and root samples of rice varieties NPB and 9522 in drought and salt stress treatments. (**b**) The Log_2_(FPKM) values for each candidate reference gene in the six RNA-seq datasets. The horizontal line in the box is the median line, the white box is the mean, and the black dot outside the box is the outlier. (**c**) Heatmap of the expression of candidate reference genes retrieved from RiceXPro data including various tissue/organs of different developmental stages and circadian time points. Blue indicates low transcript levels, and red indicates high transcript levels in log2 value.

**Figure 3 plants-12-03946-f003:**
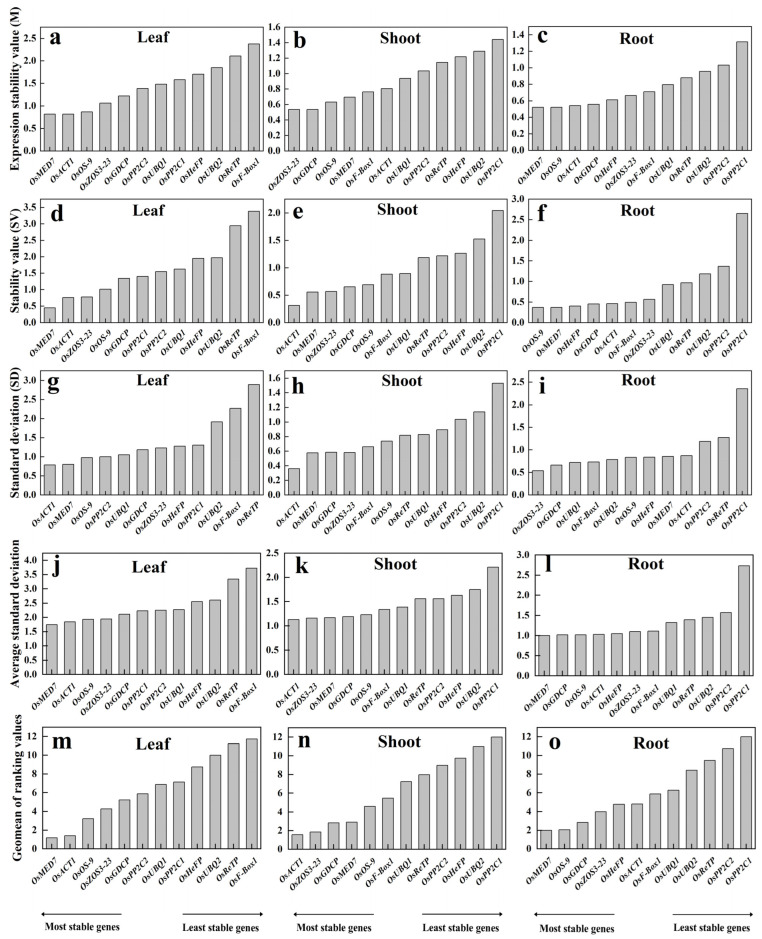
Expression stability of the 12 candidate reference genes (OsACT1, OsF-Box1, OsPP2C1, OsPP2C2, OsUBQ1, OsUBQ2, OsGDCP, OsHeFP, OsMED7, OsOS-9, OsReTP, and OsZOS3-23) by using algorithms geNorm (**a**–**c**), NormFinder (**d**–**f**), BestKeeper (**g**–**i**), ΔCt (**j**–**l**) and RefFinder (**m**–**o**).

**Figure 4 plants-12-03946-f004:**
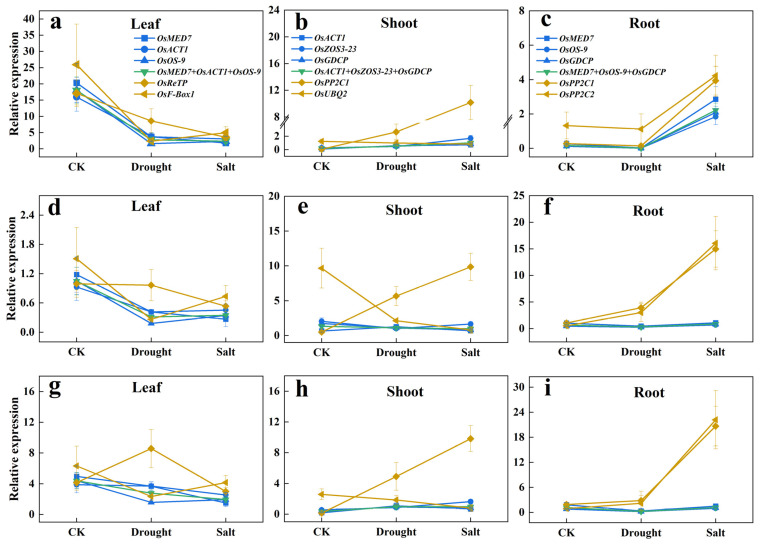
Relative expression levels of target genes in different sample sets normalized by the most or the least stable novel reference genes. (**a**–**c**) Relative expression levels of OsbZIP71 in different tissues. (**d**–**f**) Relative expression levels of OsKTN80b in different tissues. (**g**–**i**) Relative expression levels of OsNRT1.1B in different tissues. (The color legends used for the leaf, shoot, and root modules are consistent. Blue lines: normalized by each of the top three most stable candidate reference genes; Green line: normalized by combination of the top three most stable candidate reference genes; Yellow lines: normalized by each of the bottom two candidate reference genes; CK: mock treatment with water).

**Table 1 plants-12-03946-t001:** Locus and primers info of 12 candidate references and three target genes.

Gene	Locus ID	Primer Sequence	Amplicon Product Tm (°C)	Amplicon Product Size (bp)	Efficiency (%)	R^2^
*OsACT1*	LOC_Os08g28190	F: GTTCCCTTGTGTTGTTGGGC R: CCTCAAGTCAGCACAAGCCG	59	103	95.8	0.99
*OsF-Box1*	LOC_Os10g26990	F: TGTATATGATGGCAAGTG R: ATTGGATGATGGTAGGTA	56.3	82	94.6	0.99
*OsPP2C1*	LOC_Os05g51490	F: CATTGTTGTCCATCTTGTT R: CTCATCAGCACCTATCAG	59	79	100.7	0.99
*OsPP2C2*	LOC_Os06g43640	F: CAGGTTCATATCACTCAAG R: ATCATACTGGTGCTCATT	59	117	96.3	0.99
*OsUBQ1*	LOC_Os04g48770	F: ATTATTGAGAGGACTGTGAAGT R: TGATGGTTGCTGCGGATT	56.3	78	106.7	0.98
*OsUBQ2*	LOC_Os04g37950	F: ACTGCAACAGTGTCCTCCAG R: TCAAAAGGTTCTCCTCCGCA	56.3	111	95.1	0.98
*OsGDCP*	LOC_Os01g34190	F: CGAGTTCTGCTCCTCCGTAA R: CCGTCCCGTACTTGCTACAC	59	94	94.9	1.00
*OsHeFP*	LOC_Os03g38990	F: ACACACGATGTGGTTGCTCT R: CAAGGCGGAGAGGGCTATTT	56.3	151	103.7	0.99
*OsMED7*	LOC_Os04g56640	F: TTCATCTGCACCTGAACCTCC R: TCCTCTAAGCTTGGGAGCAC	61.2	96	103.4	0.99
*OsOS-9*	LOC_Os06g43710	F: CGGGAAAATCCGACAAGTCCA R: GCATCAGCATCATATTCACCCA	61.2	78	97.8	0.99
*OsReTP*	LOC_Os08g01070	F: TGGGATTAAGAGAACAAGGAG R: TGCAGTCGCAGCATCTACTT	56.3	133	105.1	0.99
*OsZOS3-23*	LOC_Os03g61640	F: TGAATGTGTCGTTAAGATCAGC R: ACACGCATAGACATGCCCAA	56.3	87	91.1	0.98
*OsbZIP71*	LOC_Os09g13570	F: TGTGTGCCCTAACTGACATCCTGA R: AAGTTATGGGTGGCTGGTTCCAT	59	133	108.0	0.99
*OskTN80b*	LOC_Os04g58130	F: TGATGCAGGGACAAAAACATGA R: CCAGGTTTTGCATCTTCCCGGGT	59	120	93.1	1.00
*OsNRT1.1B*	LOC_Os10g40600	F: GGCAGGCTCGACTACTTCTA R: AGGCGCTTCTCCTTGTAGAC	59	104	98.2	1.00

Note: The last three genes are target genes detected in the trial tests; R^2^: The linear regression correlation coefficients in actual RT-qPCR analysis.

## Data Availability

The data presented in this study are available upon request from the corresponding author.

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
