# Peer review of "Validation of Novel Reference Genes in Different Rice Plant Tissues through Mining RNA-Seq Datasets"

_plants, 2023, doi:10.3390/plants12233946_

Round 1

Reviewer 1 Report (Previous Reviewer 1)

Comments and Suggestions for Authors

As far as I'm concerned, it's ready for publication if the rest of the referees agree. 

Author Response

Thanks for the reviewer’s approval.

Reviewer 2 Report (Previous Reviewer 2)

Comments and Suggestions for Authors

  1. Thank you for reviewing manuscript-ID on date. We wonder if it is convenient for you to check the revised version on date. Alternatively, if you have been on holiday, please feel free to let us know. We will let our Academic Editor help with checking the revision . Thank you in advance. 

1.The title is clear and reflects the content of the study. However consider adding the word "novel" or "newly identified" to emphasize the contribution of the study in identifying novel reference genes.

2. Abstract: Consider briefly mentioning the significance of the study and how it addresses a gap in the current understanding of reference genes in rice. Specify the number of rice varieties included in the RNA-seq datasets in the abstract. 

3. Introduction: Provide more context on why specific tissues or growth conditions are challenging in terms of reference gene stability. Consider starting with a broader introduction to gene expression analysis before narrowing down to the specific challenges in rice. 

4. Discussion: Discuss any potential limitations of the study, such as the generalizability of findings to other rice varieties or growth conditions not covered in the datasets. Consider discussing the implications of the findings for future gene expression studies in rice and potentially other crops. Provide a more nuanced discussion on the concept of "ideal" reference genes, acknowledging the challenges in finding universally stable reference genes. 

5. Conclusion: Clearly state the practical implications of the findings for researchers conducting gene expression studies in rice. Highlight the broader applications of the RNA-seq analysis method in discovering stable reference genes beyond rice. 

6. Given the variability in growth conditions and stress responses, do you foresee the need for specific reference genes tailored to certain stress conditions, or do you believe the identified reference genes can be universally applied across a wide range of conditions? 

7. In the Discussion section, you mentioned the potential use of a curated pool for a more comprehensive search for optimal reference genes. Could you provide more insights into how such a curated pool might be established and its potential benefits? 

8. Can you elaborate on the specific criteria used to determine the stability of reference genes in the RNA-seq datasets, and how these criteria contribute to the robustness of your findings? 

9. Given the variability in growth conditions and stress responses, do you foresee the need for specific reference genes tailored to certaiin stress conditions, or do you believe the identified reference genes can be universally applied across a wide range of conditions? 

10. Considering the importance of reference genes in normalizing RT-qPCR data have you explored considered the impact of variations in primer efficiency on the accuracy of gene expression quantification? 

The research article is well-structured and presents a comprehensive study on the validation of reference genes in different rice plant tissues using RNA-seq datasets. Before finalizing the article it would be beneficial to have it reviewed by an English language expert to ensure grammatical accuracy, clarity and adherence to professional writing standards.

Author Response

Reviewer 3 Report (Previous Reviewer 3)

Comments and Suggestions for Authors

In the review process of the previous version, I raised a concern about the experimental design of this study because the authors used various kind of RNA-seq data for selecting candidate genes but used only drought-stressed plants as a sample for validation of the candidate genes.

The authors explained the aim of this study in the cover letter that they tried to find out reference genes with stable expression across different experimental conditions. Now I understand the concept, and I’m convinced with the authors’ explanation. I also found that the current manuscript is revised properly, and there are no points to be modified.

Author Response

Thanks for the reviewer’s approval.

Reviewer 4 Report (New Reviewer)

Comments and Suggestions for Authors

Dear Editor,

      Many thanks for providing me with the opportunity to review the paper titled “Validation of reference genes in different rice plant tissues through mining RNA-seq datasets” by Liu et al. In the mentioned draft the authorship team treated japonica rice cultivars with salinity (30 mmol/L NaCl) and drought stress. In the next phase of their experiment,        RNA sequencing was done on various rice tissue and some candidate reference genes were screened. I found the paper interesting and helpful, especially for rice researchers, however, I have some comments and concerns as follows:

Abstract

- I strongly believe that the material and method section in. the abstract should be organized and improved. In the abstract, the authors mentioned that the RNA-sequencing has been done on 5 different genotypes, however authors started the material and method section by mentioning “Two japonica rice cultivars”. I believe that “at first sight” would be confusing for the readers when they have a glimpse at the abstract material and method. Therefore, as mentioned, it is good if authors make the abstract more organised and in parallel with the materials and method section.

- In addition, there is not any information about the stress treatments in the abstract.

- As I found, the authors used one salinity stress treatment and one drought stress condition, however they emphasised that “their findings provide a guidance for optimizing reference gene selection and enhance the accuracy of gene expression studies in rice under diverse conditions”. My specific question is how they know that these genes are great candidates for different levels of salinity or drought stress. Which proof shows that these genes are constitutive under different levels of stress?

Materials and Methods

- As mentioned in the abstract, why do authors use “30 mmol/L NaCl for salinity and air-dry for drought stress? Why not other concentrations of NaCl? And how similar is air-dry with real drought stress that rice encounters? Basically, different concentrations of PEG show a reliable result to be used for drought stress. Can we measure the level of drought stress when the rice is under air-dry conditions?

- Generally, the important aspect of reference genes is the concept of the expression which is either constitutive or inducible. As I found the expression of some of the genes has been changed over the rive genotype and/ or type of tissue. How can the author explain different expressions and are these genes inducible or constitutive? If constitutive we can accept them as reference genes, however if they show up as inducible how the authors interpret it?

Introduction

-This section just needs some minor corrections including English polishing and writing the full name of the genes before abbreviations. In addition, it's needed if the authors add the problem statement, hypothesis and the objectives of the study in the last paragraph. It can give the readers a full insight into the workflow of the paper.    

Results and discussion

- My question is, why did the authors select these 8 genes, what was the hypothesis behind this selection? Is there any information about the function of each gene in the pathways?

Comments on the Quality of English Language

minor

Author Response

Reviewer 5 Report (Previous Reviewer 4)

Comments and Suggestions for Authors

Thank you for the opportunity to review the revised version of paper. I must say that I am impressed with the improvements which authors have made in this revision. The revised paper highlights the originality of the research and its significant contribution to the field. The findings presented offer valuable insights and extend the existing knowledge in a meaningful way. The discussion section effectively analyzes the results and their implications, highlighting the practical and theoretical contributions of the study. The writing style has been refined, resulting in a more concise and precise manuscript. The language is clear and accessible, making it easier for readers to grasp the main ideas. I commend author`s efforts in revising the paper's grammar, punctuation, and overall language usage, as it greatly enhances the overall readability and comprehension.

I am happy to recommend its acceptance in its present form.

Author Response

Thanks for the reviewer’s approval.

This manuscript is a resubmission of an earlier submission. The following is a list of the peer review reports and author responses from that submission.

Round 1

Reviewer 1 Report

Comments and Suggestions for Authors

The manuscript by Liu et al. is a technical work to improve qPCR conditions in order to have better and more reliable housekeeping genes to refer the results.

Although interesting results are presented, some major issues make it unsuitable for publication in this journal in its present form.

Regarding the grow conditions, it is not clear if they are the same for the samples analyzed in this experiment and the datasets selected to find the new reference genes.

First of all, the work is more appropriate for a technical or qPCR speciliazed journal since the results are not related to Plant biology.

Moreover, the RNAs or cDNAs used for the qPCR are not described. Are they from other published work?  Are they deposited in any repository? Since the results obtained seem to be different, it is necessary to see the previous results.

Furthermore, one of the genes was previously used in qPCR (ACT1), and the variability of other previously used genes (such as CK_1a3, eIF4, GADPH-2, UBC32, and UBQ5) could not be considered high. Thus, the improvement is not clear for the new genes proposed.

 Minor issues

LinReg analysis could be considered, at least for discussion, since it gives absolute quantity results (although by statistical methods).

Reviewer 2 Report

Comments and Suggestions for Authors

This manuscript offers a valuable contribution to the field of reverse transcription quantitative real-time PCR (RT-qPCR) in the context of rice gene expression analysis. The comprehensive use of RNA-seq datasets to identify and evaluate novel candidate reference genes tailored to specific tissue types and growth conditions is commendable. The rigorously conducted assessments of reference gene suitability across typical rice tissues, including those subjected to salt and drought stresses, have yielded robust recommendations. The findings not only enhance our understanding of reference gene selection but also provide practical guidance for optimizing such selections in diverse conditions. The quality and significance of this research make it a valuable addition to the scientific literature, and we are pleased to accept this manuscript for publication. I have happy to accept this manuscript in the current format. 

Reviewer 3 Report

Comments and Suggestions for Authors

In this study, the authors tried to identify reference genes for RT-qPCR with stable expression level in rice from RNA-seq datasets. Proper selection of reference genes is an important issue as the authors mention in the text, and the results of this study provide some useful guidance for choosing stable reference genes. However, there are several concerns regarding this study as follows.

1) First of all, the gene IDs of the candidate genes identified in this study are missing in the manuscript, which should be presented clearly.

2) The strategy of searching for candidate reference genes using transcriptome data is very excellent, but similar approach has been used in previous studies and this is not the first one in rice. The authors should cite Xu et al. (2015) (https://doi.org/10.1371/journal.pone.0142015) and Soni et al. (which is already cited as ref. 12 in the current manuscript) to explain the usefulness of transcriptome data in Introduction.

3) The authors used several RNA-seq datasets from various experiments with different treatments/tissues. However, they examined the stability of the candidate genes using test samples of drought stress treatment which are different from the treatments of RNA-seq data.

If the authors tried to identify stable reference genes in drought-stressed samples, they should search for candidate genes from transcriptome data of drought stress experiments. Is it appropriate to use different samples from RNA-seq datasets for validating the suitability? If they think so, I’d like the authors to explain the basis of their notion.

In addition, there must be a particular reason for using RNA-seq data from various kind of treatments not related to drought (CO2, nitrogen rate, and gamma-radiation), which should be explained in the text.

4) The authors examined expression stability of the commonly used reference genes in Figure 1 and mention that they are not stable genes, but is it true?

The data of Figure 1 show that some genes (ex. CK_1a.3 and GAPDH-2) are relatively stable. The authors judged that their expression are low, but FPLM values of these genes seem to be similar to those of the newly identified candidate genes shown in Figure 2b. Therefore, Figure 1a should be presented in the same scale as Figure 2b.

5) The authors claim that OsMED7 and ACT1 are the most stable genes in leaves according to Figure 3. I can agree that they are suitable at least in leaf sample, but Figure 2 indicates that their expression can be varied depending on treatments/tissues. For example, there are several data points with large variation in FPKM values for these genes in Figure 2b. From what kind of treatments/tissues are these outliers? The data of RiceXPro in Figure 2c also indicate that expression of these genes fluctuates largely. Considering this situation, it would be better to test the suitability of the candidate genes using samples from various treatments/tissues as well as RNA-seq datasets.

Comments on the Quality of English Language

The manuscript is mostly well-written, but there are several minor errors.

Reviewer 4 Report

Comments and Suggestions for Authors

Title:

The title is concise and informative, but it might consider adding the word "Validation" to emphasize that not only to identified candidate reference genes but also assessed their suitability. For example: "Validation of Reference Genes in Different Rice Plant Tissues through Mining RNA-seq Datasets."

Abstract:

In the first sentence, consider specifying "quantitative gene expression analysis" instead of just "measurement of gene expression" to provide more clarity on the technique. So it would read: "Reverse transcription quantitative real-time PCR (RT-qPCR) is arguably the most prevalent and accurate technique for quantitative gene expression analysis."

Mention the number of rice varieties from which you drew RNA-seq data. This will give readers a better sense of your dataset's diversity.

In the sentence "Comprehensive analysis of two rice varieties subjected to salt and drought stresses have revealed...," consider specifying the number of samples or replicates used in this analysis to indicate the robustness of your findings.

In the last sentence, you might clarify that these findings not only provide guidance for "optimizing reference gene selection" but also enhance the accuracy of gene expression studies in rice under diverse conditions.

Throughout the Paper:

Ensure consistency in terminology. For example, you mentioned "OsOS-9" and "OsMED7" in uppercase; ensure this consistency with all gene names, either in uppercase or in italics.

Methods:

Mention the criteria used for selecting the 12 novel candidate reference genes from the RNA-seq datasets. Was this based on expression stability, abundance, or some other metric?

Results:

Provide a brief explanation or context for why these specific genes (e.g., OsMED7, ACT1, etc.) were chosen as reference genes. What makes them suitable compared to previously used reference genes?

Figures : Figures 1-4 are very good self explanatory. 

Discussion:

Discuss the implications of your findings in the broader context of rice research. How might the identification of these new reference genes impact future studies in this field?

Conclusion:

In the conclusion section, reiterate the key takeaways and practical implications of your study, emphasizing how these newly identified reference genes can benefit researchers in the field.

References : Some references are incomplete.

Paper is recommend for publication after minor revision.